# Parameterization of downward longwave radiation based on long-term baseline surface radiation measurements in China

Junli Yang[1,2], Jianglin Hu[1,2], Qiying Chen[1,2], and Weijun Quan[3,4]

[1]CMA Earth System Modeling and Prediction Centre (CEMC), China Meteorological Administration, Beijing, 100081, China
[2]State Key Laboratory of Severe Weather (LaSW), China Meteorological Administration, Beijing, 100081, China
[3]Beijing Weather Forecast Centre, Beijing Meteorological Service, Beijing, 100089, China
[4]Environmental Meteorology Forecast Centre of Beijing-Tianjin-Hebei, Beijing, 100089, China

*Correspondence to*: Weijun Quan (quanquan78430@163.com)

**Abstract.** Downward longwave radiation (DLR) affects energy exchange between the land surface and the atmosphere, and plays an important role in weather forecasting, agricultural activities, and the development of climate models. Because DLR is seldom observed at conventional radiation stations, numerous empirical parameterizations have been presented to estimate DLR from screen-level meteorological variables. The reliability and representativeness of parameterization depend on the coefficients regressed from the simultaneous observations of DLR and meteorological variables. Only a few previous studies have attempted to build parameterizations over regions in China such as the Tibetan Plateau and East China. In this study, a long-term (2011–2022) hourly dataset of DLR and meteorological elements, obtained from seven stations of the China Baseline Surface Radiation Network, was used to recalculate the coefficients of the Brunt and Weng models, and to develop a new model. Results showed that the mean bias error (MBE) and relative MBE (rMBE) between the measured clear-sky DLR and that estimated using the Brunt, Weng, and new models were −4.3, −5.1, and 3.7 W m$^{-2}$ and −1.5 %, −1.8 %, and 1.3 %, respectively. The root mean squared errors (RMSEs) where in the range of 13.8–14.3 W m$^{-2}$ and the relative RMSEs (rRMSEs) were approximately 5.0 %. The MBEs (rMBEs) of the Brunt, Weng, and new models under all-sky conditions were −2.8 W m$^{-2}$ (−1.0 %), −6.1 W m$^{-2}$ (−2.1 %), and −1.5 W m$^{-2}$ (−0.5 %), respectively. The RMSE (rRMSE) of the parameterization models in retrieving all-sky DLR was ~17.5 W m$^{-2}$ (~6.1 %). Therefore, the models are considered suitable for retrieval of DLR over China.

## 1 Introduction

Downward longwave radiation (DLR) on the ground is one of the fluxes involved in the exchange of energy between Earth's surface and the atmosphere (e.g., Idso and Jackson, 1969; Konzelmann et al., 1994; Gabathuler et al., 2001; Sridhar and Elliot, 2002). Consequently, DLR plays a vital role in weather forecasting, agricultural production (e.g., prediction of frost

and crop temperature), climate simulations, and water cycle modeling (e.g., Crawford and Duchon, 1999; Wild and Cechet, 2002; Bilbao and De Miguel, 2007; Li et al., 2017).

In comparison with other radiation components, DLR is seldom observed at conventional radiation stations (e.g., Iziomon et al., 2003; Stephens et al., 2012). Therefore, considerable effort has been made to develop simple parameterization methods to calculate DLR from easily measured meteorological variables (Duarte et al., 2006). As identified by Ångström (1915), clear-sky DLR can be determined from the emissivity and effective temperature of the atmosphere. Under clear-sky conditions, as much as 60 % (90 %) of atmospheric emission is derived from the atmosphere within the first 100 m (1 km). When the sky is overcast, more than 90 % originates from within first 1-km layer between the ground and the bottom of the cloud (Ohmura, 2001). Following the pioneering work of Ångström, numerous investigators have presented empirical relationships between effective atmospheric emissivity (hereafter referred to as emissivity) under clear-sky conditions and vapor pressure ($e$) (e.g., Brunt, 1932; Weng et al., 1993; Niemelä et al., 2001). Nevertheless, for a limited isothermal atmosphere emissivity would be less than unity and independent of temperature only if the atmosphere were of a constant greyness. In the real atmosphere, emissivity must, in principle, be temperature ($T_a$) dependent (e.g., Swinbank, 1963; Idso and Jackson, 1969). Moreover, some investigators even pointed out that the empirical emissivity depends on the dewpoint temperature ($T_d$) (e.g., Berdahl and Fromberg, 1982), $e$ and $T_a$ (e.g., Brutsaert, 1975; Satterlund, 1979; Idso, 1981; Iziomon et al., 2003), relative humidity ($\phi$) and $T_a$ (e.g., Carmona et al., 2014) or even the total amount of water vapor (e.g., Ruckstuhl et al., 2007). Under all-sky conditions, the presence of clouds can increase emissivity and atmospheric radiation. Clouds generally consist of water vapor, water droplets, or ice crystals. They absorb thermal radiation very strongly and radiate similar to a black body in the infrared range (Heitor et al., 1991). Many studies have demonstrated that all-sky emissivity can be well predicted from clear-sky emissivity with correction for cloud effects (e.g., Crawford and Duchon, 1999; Bilbao and De Miguel, 2007; Wang and Liang, 2009; Alados, et al., 2012; Li et al., 2017; Liu et al., 2020).

Because the regression coefficients of empirical parameterization models exhibit spatial dependence (e.g., Goss and Brooks, 1956; Brutsaert, 1975; Marthews et al., 2012; Liu et al., 2020), they should be recalculated on the basis of observations over wider regions to ensure their accuracy and representativeness in estimating DLR. For instance, Wang and Liang (2009) assessed the performance of clear-sky DLR parameterization models presented by Brunt (1932) and Brutsaert (1975) at 36 global sites. However, owing to the shortage of high-quality DLR measurements in China, most previous works focused on retrieval of DLR over only a few regions, e.g., the Tibetan Plateau (e.g., Weng et al., 1993; Zhu et al., 2017; Liu et al., 2020) and East China (Wang and Liang, 2009). Therefore, these models might not represent optimal parameterizations suited to retrieval of DLR over other areas of China. Fortunately, the China Meteorological Administration has established the China Baseline Surface Radiation Network (CBSRN) in 2007 (Li et al., 2013), which currently comprises seven stations (Mohe, Xilinhot, Yanqi, Shangdianzi, Xuchang, Wenjiang, and Dali). Nine radiometric components including DLR are measured at 1-min intervals at CBSRN stations. The purpose of this study was to recalculate the coefficients of the Brunt (1932) model and the Weng (1993) model, and to develop a new parametric formula based on a long-term (2011–2022) hourly dataset obtained from the CBSRN stations.

## 2 Site, instruments, and data

### 2.1 Site description

Figure 1 shows the geographical locations of the seven CBSRN stations: Mohe (MH; 52.97 °N, 122.52 °E; 438.5 m a.s.l.), Xilinhot (XL; 44.13 °N, 116.33 °E; 1003.0 m a.s.l.), Yanqi (YQ; 42.05 °N, 86.61 °E; 1056.5 m a.s.l.), Shangdianzi (SDZ; 40.65 °N, 117.12 °E; 293.3 m a.s.l.), Xuchang (XC; 34.07 °N, 113.93 °E; 67.2 m a.s.l.), Wenjiang (WJ; 30.75 °N, 103.86 °E; 547.7 m a.s.l.), and Dali (DL; 25.71 °N, 100.18 °E; 1990.5 m a.s.l.). It can be seen from Table 1 that these stations are distributed in seven representative climatic zones, i.e., the cold temperate zone (MH), middle temperate semiarid zone (XL), middle temperate arid zone (YQ), warm temperature semihumid zone (SDZ), northern subtropical humid zone (XC), middle subtropical humid zone (WJ), and subtropical humid zone (DL). Additionally, the elevation of three stations (i.e., XC, SDZ, and MH) is <500 m a.s.l., one station (WJ) has medium elevation (547.7 m a.s.l.), and the other three stations (i.e., XL, YQ, and DL) have elevation >1000 m a.s.l. (Table 1). Moreover, the CBSRN stations represent various land covers in China. For instance, MH is the northernmost meteorological station in China surrounded by forest, which is located in the northwestern suburbs of Mohe County, Heilongjiang province (Liu et al., 2018). XL lies on the central of Inner Mongolia, where the main land cover is steppe. YQ, located on the northern margin of the Tarim Basin, is one of the representative stations in the desert and Gobi in northwest of China. SDZ is located in the northern North China Plain and only a few small villages with a sparse population around it (Zhou et al., 2021). XC is located in the central Henan province, which is surrounded by a wheat field and become one of typical representative stations for farmland in China. WJ is located in Sichuan Basin, which represents a paddy field. As a part of the Dali National Climate Observatory near the Erhai Lake in Yunnan province, DL is a represent station for wetlands.

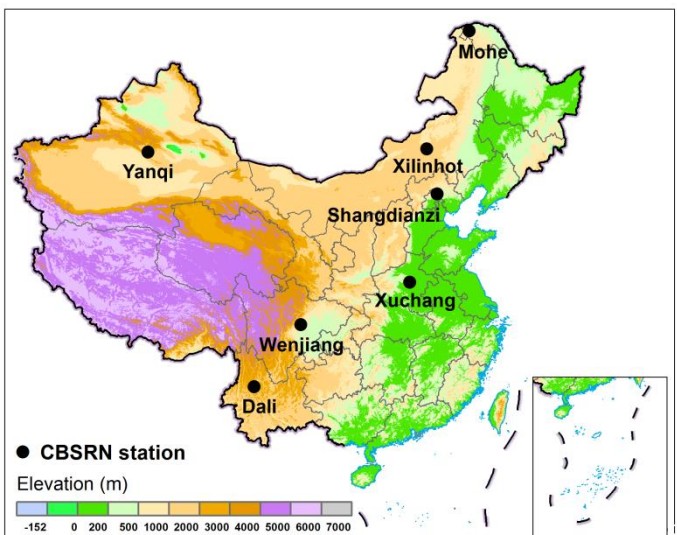

**Figure 1.** Geographical locations of seven CBSRN stations in China. Terrain data represent GTOP30 digital elevation model (ftp://edcftp.cr.usgs.gov/data/gtopo30/global/).**Table 1.** Basic descriptions of CBSRN stations in China.

| Station name | Station ID | Latitude (° N) | Longitude (° E) | Altitude (m a.s.l) | Climatic zone | Measure period |
|---|---|---|---|---|---|---|
| Mohe | 50136 | 52.97 | 122.52 | 438.5 | Cold temperate zone | Jan 2013 − present |
| Xilinhot | 54102 | 44.13 | 116.33 | 1107.0 | Middle temperate semiarid zone | Jun 2007 − present |
| Yanqi | 51567 | 42.05 | 86.61 | 1056.5 | Middle temperate arid zone | Jan 2013 − present |
| Shangdianzi | 54421 | 40.65 | 117.12 | 293.3 | Warm temperate semihumid zone | Jan 2013 − present |
| Xuchang | 57089 | 34.07 | 113.93 | 67.2 | Northern subtropical humid zone | Jan 2013 − present |
| Wenjiang | 56187 | 30.75 | 103.86 | 547.7 | Middle subtropical humid zone | Jan 2013 − present |
| Dali | 56751 | 25.71 | 100.18 | 1990.5 | Subtropical humid zone | Jan 2013− present |

## 2.2 Instruments and data

The CBSRN stations use IR02 pyrgeometers (Huksflux, the Netherlands) to measure DLR. The spectral range of the IR02 instrument is 4.5–42 μm, which covers most of spectral range of atmospheric longwave radiation, making it a suitable instrument for measuring DLR in most cases. Moreover, to avoid influences from solar radiation, the IR02 is shaded by a ball mounted on a FS-ST22 automatic solar tracker (Jiangsu Radio Science Research Institute Co. Ltd., China) during observation. Its temperature dependence is within ±3 % (−10 to 40 ℃), and a ventilation/heating system is installed to reduce the influence of environmental temperature and to prevent dew/dust fall on its window. Note that the field of view (FOV) of the IR02 instrument is 150 ° rather than the desired 180 °, which means its price is attractive, while the accuracy loss is relatively minor (Hukseflux, 2022). The IR02 sampling frequency is 1 Hz and the 1-min averaged data are stored using a WUSH-BR data logger (Jiangsu Radio Science Research Institute Co. Ltd., China). To assure the DLR measured at CBSRN is traceable to the World Radiometric Reference like that observed at Baseline Surface Radiation Network (Driemel et al., 2018), the IR02 pyrgeometers used in this study were calibrated against the reference CGR4 pyrgeometer (Kipp & zonen, the Netherlands) of China Meteorological Administration (CMA). The CGR4 can be traced to the World Infrared Standard by participating in the International Pyrgeometer Comparison organised by Physikalisch-Meteorologisches Observatorium Davos and World Radiation Center (e.g., Gröbner et al., 2014; PMOD/WRC, 2022).

A fisheye camera, mounted on top of the HY-WP1A Intelligent Weather Observation System (Huayun Sounding Meteorological Technology Inc., China), is used to automatically record CF data. Full-sky photographs with a FOV of 180 ° are acquired at 1-min intervals. The photographs are then processed using artificial intelligence image detecting technology to yield hourly CF data with uncertain <10 % (Hua et al., 2021).

Meteorological elements (i.e., $T_a$, $e$, and $\phi$) are observed by an automatic weather station (AWS) at 1-min intervals and the data are stored using a HY3000 data logger (Huayun Sounding Meteorological Technology Inc., China).

The data used in this study, which were downloaded from the China Meteorological Administration Data Service (http://idata.cma/cmadaas/), undergo strict quality controls by meteorological experts and trained engineers of the National Meteorological Information Centre of China. Note that the DLR data measured by the IR02 instruments at high-elevation stations (i.e., MH and YQ) under extremely dry and cold synoptic conditions, in which irrational DLR measurements might be produced due to the high  temperature dependency of the IR02 pyrgeometer (Hukseflux, 2022), were not involved in this study.

## 3 Methods

### 3.1 Emissivity calculation

Effective atmospheric emissivity (ε) is defined as the ratio of incoming long-wave radiation to blackbody radiation at screen-level air temperature (e.g., Monteith, 1961; Rodgers, 1967; Prata, 1996):

$$\varepsilon = \frac{DLR}{\sigma T_a^4}, \tag{1}$$

where DLR is the downward hemispheric longwave irradiance (W m$^{-2}$) at the ground, which can be observed directly by a pyrgeometer; $T_a$ is the screen air temperature (K) measured by the AWS; and σ is the Stefan–Boltzmann constant (5.6697×10$^{-8}$ W m$^{-2}$ K$^{-4}$).

### 3.2 Statistical methods

This study used the nonlinear curve fitting method, orthogonal distance regression (ODR) iteration algorithm and Levenberg–Marquardt iteration algorithm to regress the coefficients of the parameterization models. Parameterizations were assessed by means of statistical parameters such as mean bias error (MBE), relative MBE (rMBE), root mean squared error (RMSE), relative RMSE (rRMSE), and the correlation coefficient (*r*). The MBE is an indicator adopted to denote whether predictions from the parameterization are overestimates (positive values) or underestimates (negative values) in comparison with the measurements. The RMSE accounts for the average magnitude of the errors but it does not provide an indication of the direction of the errors. The correlation coefficient *r* reflects the linear agreement between the observed parameter and the estimated variable (e.g., Gubler et al., 2012; Zhou et al., 2021).

## 4 Results

### 4.1 Clear-sky emissivity parameterization

In this study, the coefficients of the Brunt model and the Weng model were calibrated using the nonlinear curve fitting method with 12,368 hourly data pairs (DLR and *e*) under clear-sky condition (defined as the corresponding cloud fraction equal to zero) observed at seven CBSRN stations between January 2011 and December 2017. The Brunt model is one of the

earliest pronounced models, in which a simple formula connecting the downward radiation from the atmosphere, the total black-body radiation at temperature, and the vapour pressure (Brunt, 1932). The Weng model is one of the earliest parameterization presented to retrieve the DLR over China area from the atmospheric temperature and vapour pressure based on the experimental observation data on the Tibetan Plateau (Weng et al., 1993). Note that both the Brunt model and the Weng model are single-parameter parameterization models because only one parameter ($e$) is adopted as input in these models. The Brunt model is a power function of $e$ with an exponent of 1/2, whereas the Weng model is a natural logarithm function of $e$. A two-parameter model such as the Brutsaert (1975) model, in which $e$ and $T_a$ are both used as input parameters, is recognized to be more reasonable than a single-parameter model in terms of the physical mechanism, especially under warm and wet conditions (e.g., Culf and Gash, 1993; Prata, 1996). In this study, we developed a two-parameter parametric formula (hereafter referred to as the parametric formula) that is similar to the Brutsaert model except the exponent of the function is set to 1/3 rather than 1/7. The coefficients of the parametric formulae were computed on the basis of the clear-sky hourly dataset (DLR, $e$, and $T_a$) using the nonlinear curve fitting method together with the ODR iteration algorithm.

The formulae of the parameterization models for retrieving clear-sky emissivity can be expressed as follows:

$$\varepsilon_{clr,B} = 0.599 + 0.053\sqrt{e}\,, \tag{2}$$

$$\varepsilon_{clr,W} = 0.590 + 0.075\ln(1+e)\,, \tag{3}$$

$$\varepsilon_{clr,Y} = 0.532 + 0.808\sqrt[3]{e/T_a}, \tag{4}$$

where $\varepsilon_{clr,B}$, $\varepsilon_{clr,W}$, and $\varepsilon_{clr,Y}$ represent the clear-sky emissivity retrieved from the Brunt model, Weng model, and new model developed in this study, respectively, $e$ (hPa) is vapor pressure, and $T_a$ (K) is screen-level air temperature. The coefficients of determination ($R^2$) of Eqs. (2)–(4) were 0.999, 0.999, and 0.930, respectively.

The Brunt model (denoted by the black thick curve in Fig. 2a) can well fit all data pairs under most cases ($0 < e \leq 45$ hPa), especially those data pairs observed at low-elevation (<1000 m) stations such as XC (67.2 m a.s.l.) and SDZ (293.3 m a.s.l.), whereas the Weng model (denoted by the red thick curve in Fig. 2a) appears to fit the data pairs better than the Brunt model under dry conditions ($e \leq 17.5$ hPa). Note that the Weng model was proposed in terms of radiation data observed over the Tibetan Plateau, where the atmospheric vapor pressure is lower than that in other regions in China. Therefore, it can be inferred that the Weng model is suitable for estimating clear-sky emissivity over arid regions. The parametric formula developed in this study (denoted by the red thick curve in Fig. 2b) fitted the data pairs reasonably and was considered to be based on physics because it uses both $e$ and $T_a$ as input.

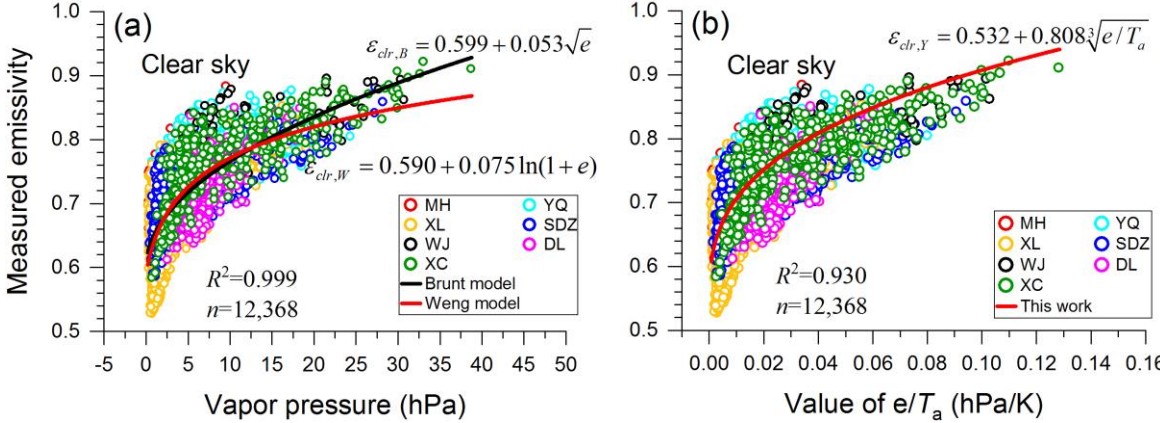

**Figure 2.** Scatter plots of measured clear-sky emissivity versus coincident measurements of (a) vapor pressure and (b) the ratio of vapor pressure to screen-level air temperature. Circles represent hourly data pairs observed at seven CBSRN stations. Black and red thick curves in (a) denote the Brunt model (Eq. 2) and the Weng model (Eq. 3), respectively. Red thick curve in (b) denotes the parametric formula developed in this study (Eq. 4).

The coefficients of the well-known Brunt model reported by previous authors as well as those derived in this study are listed in Table 2. The intercept (0.599) and slope (0.053) in the Brunt model derived in this study (Eq. 4) are consistent with those presented both by Li et al. (2017) and by Wang and Liang (2009), but differ from those provided by other investigators. Discrepancies in the coefficients might result from different atmospheric conditions (e.g., water vapor content, CF, and temperature profiles) and the temporal resolution (hourly, daily, or monthly) of the data used in establishing the parameterization models. The greater the values of slope, the larger the dependences of the parameterization formula on water vapor (Iziomon et al., 2003). Note that the value of slope greater than 0.05 in the Brunt model, Monteith (1961) model, Iziomon model, Berdahl and Martin (1984) model, Li model, and the model developed in this study, which means that these models have greater sensitivity to water vapor in comparison with other models.

**Table 2.** Coefficients of the Brunt model reported by previous investigators as well as those derived in this study. The temporal resolution of the data used to derive the coefficients and the details of network/sites at which the observations were performed are also listed.

| Reference | Network/Site | Number of Sites | Elevation(m) | Intercept | Slope | Resolution | Country |
|---|---|---|---|---|---|---|---|
| Brunt [1932] | Benson | 1 | 6 | 0.520 | 0.065 | Monthly | UK |
| Anderson [1954] | Laker Hefner | 1 | 369 | 0.680 | 0.036 | Monthly | USA |
| Goss and Brooks [1956] | Davis | 1 | 14 | 0.660 | 0.039 | Monthly | USA |
| DeCoster and Schuepp [1957] | Kinshasa | 1 | 321 | 0.645 | 0.048 | Daily | Zaire |
| Monteith [1961] | Kew | 1 | – | 0.530 | 0.065 | Hourly | England |
| Swinbank [1963] | Aspendale, Kerang | 3 | – | 0.640 | 0.037 | Hourly | Australia |

| | Diamantina | | | | | | |
|---|---|---|---|---|---|---|---|
| Berger et al. [1984] | Carpentras | 1 | – | 0.660 | 0.040 | Hourly | France |
| Berdahl and Martin [1984] | Tucson, Gaithersburg, San Antornio, Boulder St. Louis, West palm beach | 6 | – | 0.564 | 0.059 | Hourly | USA |
| Heitor et al. [1991] | Sacavem | 1 | – | 0.590 | 0.044 | Hourly | Portugal |
| Iziomon et al.[2003] | Bremgarten, Feldberg | 2 | 212, 1489 | 0.600 | 0.064 | Hourly | Germany |
| Wang and Liang [2009] | SURFRAD AsiaFlux FLUXNET AmeriFlux GAME AAN | 36 | 98–4700 | 0.605 | 0.048 | Hourly | USA Indonesia Japan China Thailand Australia Botswana Canada Germany |
| Li et al. [2017] | SURFRAD | 7 | 98–1689 | 0.598 | 0.057 | Hourly | USA |
| Liu et al. [2020] | Naqu, Nyingchi, Ali | 3 | 2290–4507 | 0.560 | 0.070 | Minute | China |
| This work | CBSRN | 7 | 67–1991 | 0.599 | 0.053 | Hourly | China |

## 4.2 All-sky emissivity parameterization

Under all-sky conditions, the emission from clouds can supplement the radiation emitted by water vapor and other gases in the lower atmosphere. Therefore, the effective emissivity of the atmosphere is higher under all-sky condition compared to that under clear-sky condition (e.g., Li et al., 2017). Numerous formulae were presented to estimate the emissivity under all-sky condition based on the emissivity parameterization under clear-sky condition and cloud fraction (e.g., Maykut and Church, 1973; Crawford and Duchon, 1999; Duarte et al., 2006; Choi et al., 2008). The formula of Duarte et al. (2006) with an adjustment of atmospheric humidity was adopted in this study. For a site like Barrow, Alaska, where both the temperature and the partial pressure of water vapor are low during much of year, the effect of atmospheric humidity on emissivity under all-sky condition can be neglected (e.g., Maykut and Church, 1973). However, the temperature and atmospheric humidity over the CBSRN stations vary over a wide range during a year, the addition of moisture correction to the formula, thus, seems more reasonable. The structure of formula to estimate the emissivity under all-sky condition in this study as:

$$\varepsilon_{all} = \varepsilon_{clr}\left(1 - \alpha CF^{\beta}\right) + \gamma CF^{\delta}\emptyset^{\zeta}, \tag{5}$$

where $\varepsilon_{all}$ represent all-sky emissivity; $\varepsilon_{clr}$ is the clear-sky emissivity calculated using Eqs. (2)–(4); CF is the cloud fraction (0–1); $\emptyset$ is relative humidity (%); $\alpha, \beta, \gamma, \delta,$ and $\zeta$ are regression coefficients, which were derived using the dataset of observations recorded at seven CBSRN stations between January 2011 and December 2020. The dataset comprises 71,204 hourly measurements of DLR, $e$, $T_a$, CF, and $\emptyset$ under all-sky conditions. The formulae derived for all-sky emissivity are as follows:

$$\varepsilon_{all,B} = \varepsilon_{clr,B}(1 - 0.178CF^{0.339}) + 0.075CF^{0.395}\phi^{0.253}, \tag{6}$$

$$\varepsilon_{all,W} = \varepsilon_{clr,W}(1 + 0.186CF^{0.499}) - 0.298CF^{0.424}\phi^{-0.360}, \tag{7}$$

$$\varepsilon_{all,Y} = \varepsilon_{clr,Y}(1 - 0.201CF^{0.796}) + 0.088CF^{1.038}\phi^{0.221}. \tag{8}$$

where $\varepsilon_{all,B}$, $\varepsilon_{all,W}$, and $\varepsilon_{all,Y}$ represent all-sky emissivity retrieved from the Brunt model, Weng model, and new model developed in this study, respectively; $\varepsilon_{clr,B}$, $\varepsilon_{clr,W}$, and $\varepsilon_{clr,Y}$ are clear-sky emissivity calculated using Eqs. (2)–(4); the coefficients of determination for Eqs. (6)–(8) are 0.745, 0.748, and 0.750, respectively.

**4.3 Emissivity validation**

To verify the clear-sky emissivity parameterization models (Eqs. (2)–(4)) defined in section 4.1, this study used an independent clear-sky dataset comprising 1,706 hourly clear-sky measurements of DLR, $e$, and $T_a$ at four CBSRN stations (YQ, XL, SDZ, and XC) acquired between January 2018 and July 2021. The MBEs (rMBEs) between the measured clear-sky emissivity and that estimated by the Brunt model, the Weng model, and model developed in this study were −0.013 (−1.8 %), −0.015 (−2.1 %), and 0.007 (1.0 %), respectively (Fig. 3a–c). The small positive MBE of the model developed in 210 this study might be attributable to the fact that two parameters ($e$ and $T_a$) are involved in the equation. Meanwhile, all models yielded analogous RMSEs (~0.039) and rRMSEs (~5.3 %), which might be a reflection of the dataset and selected independent variables used to establish the formulae. For example, the effects of $CO_2$, $O_3$, and aerosols on emissivity were not considered in these formulae (e.g., Staley and Jurica, 1972; Kjaersgaard et al., 2007; Gubler et al., 2012).

The parameterization models used to estimate all-sky emissivity (Eqs. (6)–(8)) were validated on the basis of an 215 independent dataset comprising 20,970 hourly all-sky measurements (DLR, $e$, $T_a$, CF, and $\phi$) acquired at three CBSRN stations (XL, SDZ, XC) between January 2021 and April 2022. The MBEs (rMBEs) between the measured all-sky emissivity and that calculated by the Brunt model, the Weng model, and model developed in this study were −0.006 (−0.8 %), −0.017 (−2.2 %), and −0.004 (−0.5 %), respectively (Fig. 3d–f). Note that the MBEs (rMBEs) of the all-sky emissivity were close to or even less than those of the clear-sky emissivity. One possible reason is that more samples (20,970) 220 were adopted in verifying the all-sky emissivity than were adopted in validating the clear-sky emissivity (i.e., 1,706). Another reason is that more input parameters (e.g., CF and RH) other than $e$ and $T_a$ were included in the all-sky emissivity formulae, which alleviated the abrupt variations of $e$ or $T_a$. However, the RMSE of the all-sky emissivity parameterization model was ~0.049, which is higher than that (~0.039) of the clear-sky emissivity model.

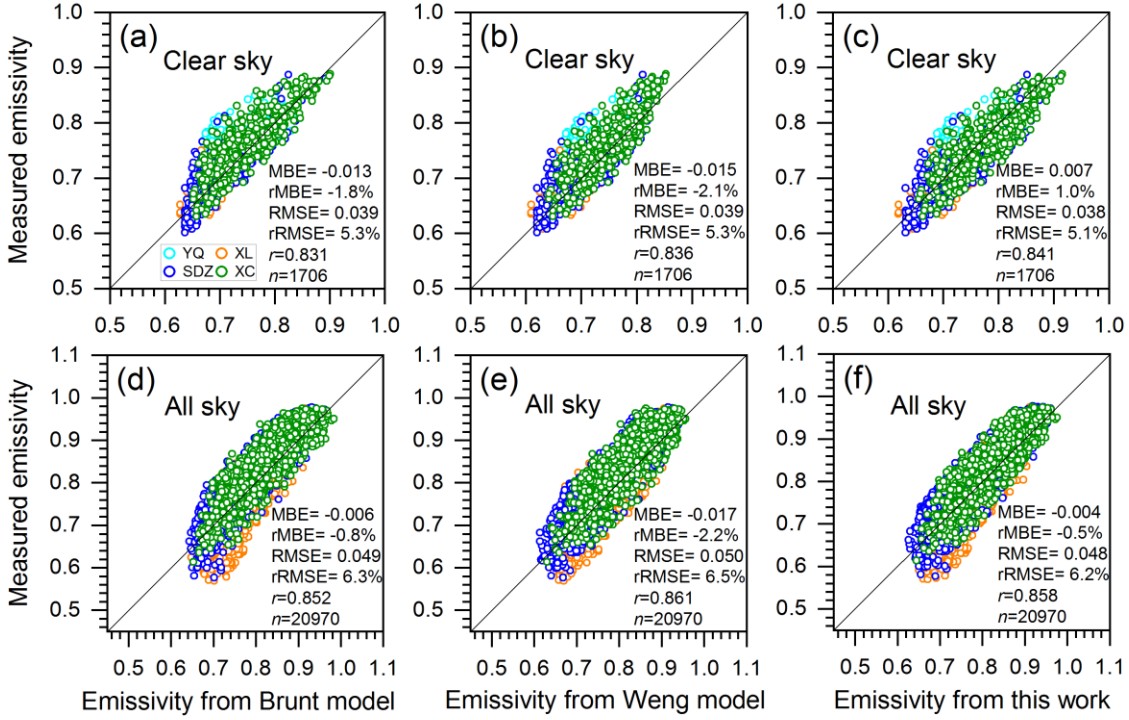

**Figure 3.** Measurements versus calculations of effective atmospheric emissivity by the Brunt model, the Weng model, and the model developed in this study for (a)–(c) clear-sky and (d)–(f) all-sky conditions. Black lines denote the 1:1 line.

To illustrate the performance of each of the parameterization models in estimating both clear- and all-sky emissivity in different seasons, several statistics are summarized in Table 3 and plotted in Fig. 4. The model developed in this study can overestimate clear-sky emissivity in all seasons except winter (with rRMBE of −1.1 %), whereas, both the Brunt model and the Weng model underestimates clear-sky emissivity in all seasons (Fig. 4a). The influence of involving $T_a$ in the model would be more noteworthy during summer and winter because $T_a$ reaches its maximum and minimum value in these seasons, respectively. Furthermore, all parametrization models exhibited apparent negative rMBEs in winter, unlike in other seasons. In winter, the $CO_2$ content over China usually reaches its annual maximum (Fang et al., 2014). Therefore, underestimation of clear-sky emissivity using parameterization models would be greater in winter owing to the effect of neglecting $CO_2$ in the models. Under all-sky conditions, the rMBEs of the parameterization models were negative in all seasons, except for the Brunt model (rMBE of 0.6 %) and the model developed in this study (rMBE of 1.5 %) in autumn (Figure 4b).

For clear-sky emissivity (Fig. 4c), the rRMSEs between the measurements and the estimations of three models were ~5 % in spring (March–May), summer (June–August), and autumn (September–November), but >5.7 % in winter (December–February). For all-sky emissivity, the rRMSEs between the measurements and the estimations of three models were closer with values of ~6.5 %, ~5.0 %, ~6.3 %, and ~7.4 % in spring, summer, autumn, and winter, respectively (Fig. 4d, Table 3).

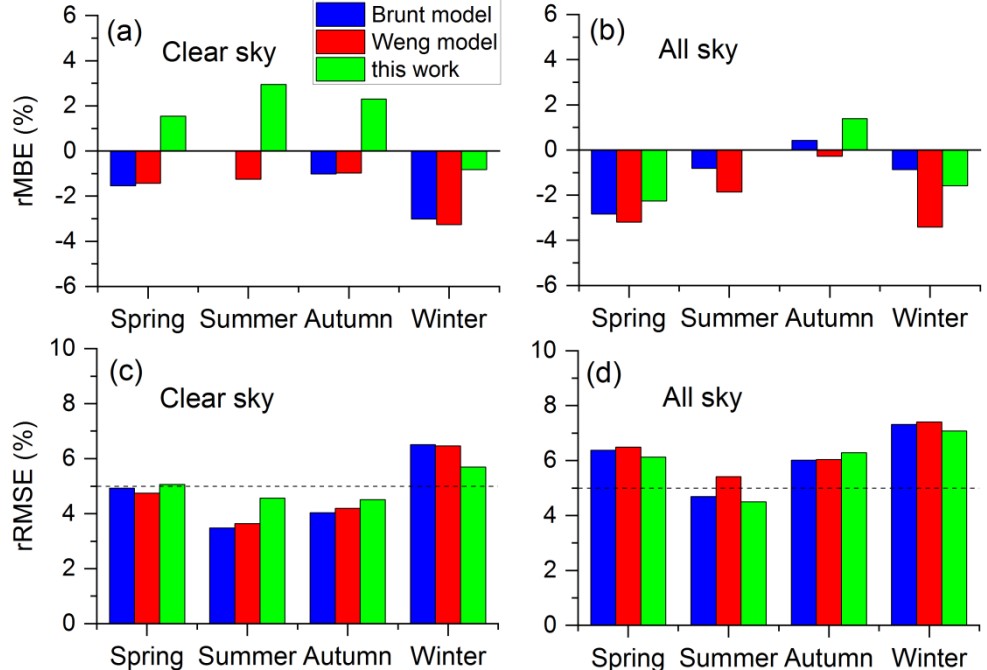

**Figure 4.** Seasonal statistics of rMBEs for three parameterization models (a) under clear-sky conditions and (b) under all-sky conditions; the corresponding rRMSEs for three models (c) under clear-sky conditions and (d) under all-sky conditions. Dashed line denotes the rRMSE value of 5 %.

**Table 3.** Comparison between the measured emissivity and those estimated using three models in four seasons under clear- and all-sky conditions.

| Season | Sky condition | Model | MBE | rMBE (%) | RMSE | rRMSE (%) | *r* | Sample number |
|---|---|---|---|---|---|---|---|---|
| Spring | Clear sky | Brunt | −0.012 | −1.7 | 0.038 | 5.1 | 0.762 | 445 |
| | | Weng | −0.012 | −1.5 | 0.037 | 4.9 | 0.772 | |
| | | This work | 0.011 | 1.4 | 0.038 | 5.0 | 0.772 | |
| | All sky | Brunt | −0.022 | −2.8 | 0.051 | 6.5 | 0.815 | 4389 |
| | | Weng | −0.025 | −3.2 | 0.052 | 6.6 | 0.819 | |
| | | This work | −0.018 | −2.3 | 0.049 | 6.3 | 0.816 | |
| Summer | Clear sky | Brunt | −0.001 | −0.1 | 0.029 | 3.6 | 0.769 | 254 |
| | | Weng | −0.011 | −1.3 | 0.030 | 3.7 | 0.750 | |
| | | This work | 0.023 | 2.9 | 0.037 | 4.6 | 0.769 | |
| | All sky | Brunt | −0.007 | −0.9 | 0.041 | 4.8 | 0.810 | 5036 |
| | | Weng | −0.016 | −1.9 | 0.047 | 5.5 | 0.763 | |
| | | This work | 0.000 | −0.1 | 0.040 | 4.6 | 0.822 | |
| Autumn | Clear sky | Brunt | −0.008 | −1.1 | 0.032 | 4.2 | 0.841 | 369 |
| | | Weng | −0.008 | −1.0 | 0.033 | 4.3 | 0.841 | |
| | | This work | 0.017 | 2.2 | 0.035 | 4.5 | 0.848 | |
| | All sky | Brunt | 0.005 | 0.6 | 0.049 | 6.3 | 0.873 | 5106 |
| | | Weng | −0.001 | −0.2 | 0.048 | 6.1 | 0.886 | |

| | | | | | | | | |
|---|---|---|---|---|---|---|---|---|
| | | This work | 0.012 | 1.5 | 0.050 | 6.5 | 0.878 | |
| Winter | Clear sky | Brunt | −0.022 | −3.1 | 0.047 | 6.6 | 0.670 | 638 |
| | | Weng | −0.024 | −3.5 | 0.047 | 6.6 | 0.688 | |
| | | This work | −0.007 | −1.1 | 0.040 | 5.7 | 0.695 | |
| | All sky | Brunt | −0.003 | −0.5 | 0.054 | 7.6 | 0.592 | 6439 |
| | | Weng | −0.024 | −3.4 | 0.054 | 7.5 | 0.709 | |
| | | This work | −0.010 | −1.4 | 0.052 | 7.2 | 0.657 | |

## 4.4 DLR validation

Based on the Eq. (1), clear-sky DLR can be calculated in terms of the measurements of screen-level temperature and the corresponding emissivity estimated using the parameterization models. Statistical results (Table 3) indicated that the MBEs (rMBEs) between the measured hourly clear-sky DLR at YQ, XL, SDZ, and XC during 2018−2021, and that estimated using the Brunt model, the Weng model, and model developed in this study were −4.3 W m$^{-2}$ (−1.5 %), −5.1 W m$^{-2}$ (−1.8 %), and 3.7 W m$^{-2}$ (1.3 %), respectively (Fig. 5a−c). The RMSE (rRMSE) of both the Brunt model and the Weng model was 13.8 W

m$^{-2}$ (4.9 %), i.e., slightly lower than the one of the model developed in this study (RMSE: 14.3 W m$^{-2}$, rRMSE: 5.1 %).

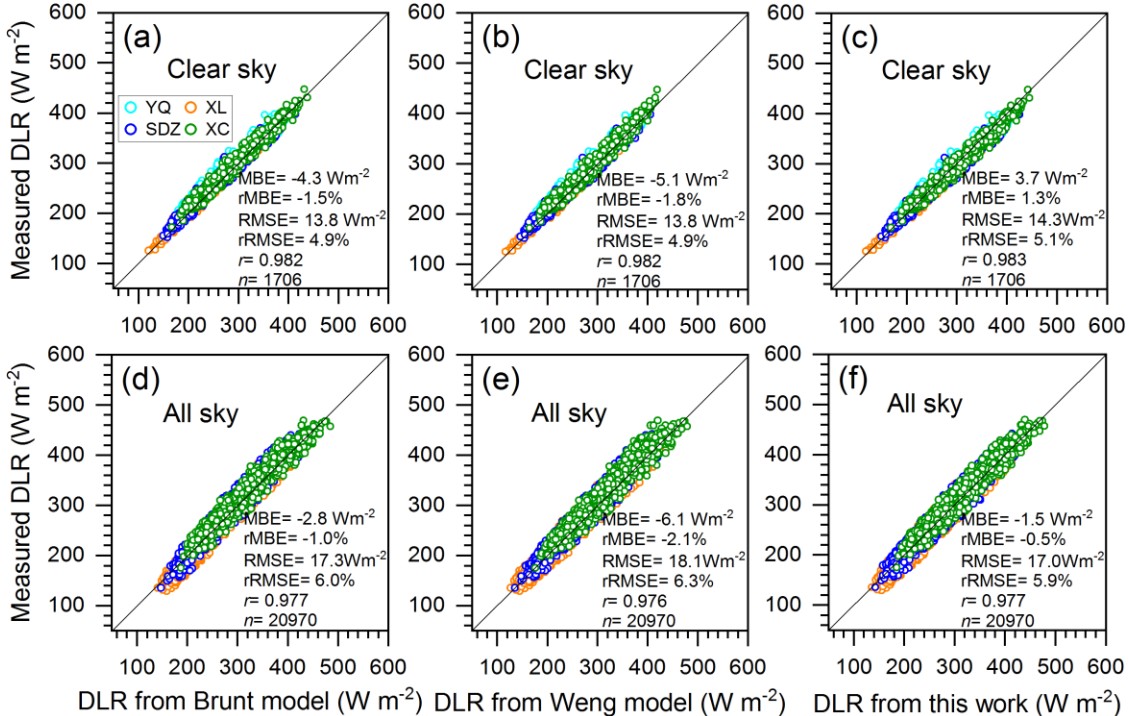

**Figure 5.** Same as Fig. 3 except for the DLR.

    Under all-sky conditions, the MBEs (rMBEs) between the measured hourly all-sky DLR at XL, SDZ, and XC from January 2021 to April 2022, and that estimated using the Brunt model, the Weng model, and parametric formula developed

in this study were −2.8 W m$^{-2}$ (−1.0 %), −6.1 W m$^{-2}$ (−2.1 %), and −1.5 W m$^{-2}$ (−0.5 %), respectively (Fig. 5d−f). The

RMSEs (rRMSEs) between the measured all-sky DLR and that retrieved using the Brunt model, the Weng model, and model developed in this study were 17.3 W m$^{-2}$ (6.0 %), 18.1 W m$^{-2}$ (6.3 %), and 17.0 W m$^{-2}$ (5.9 %), respectively. It can be seen that the RMSE (rRMSE) of the all-sky DLR retrieved from the parameterization models was ~3.5 W m$^{-2}$ (1.0 %) greater than the one of the clear-sky DLR retrieval. Occurrence of clouds and wider ranges of temperature and humidity under all-sky

conditions would disperse the relationship between the observations and the predictions of DLR.

It can be seen from Table 4 that both the Brunt model and the Weng model tend to underestimate clear-sky DLR in all seasons (MBEs in the range of −7.1 to 0.0 W m$^{-2}$), while the model developed in this study tend to overestimate clear-sky DLR in all seasons except winter (MBE of −1.8 W m$^{-2}$). Additionally, all models tend to underestimate all-sky DLR (MBEs in the range of −9.4 to −0.1 W m$^{-2}$) in all cases, except the Brunt model and the model developed in this study tend to

overestimate all-sky DLR in autumn (MBE of 1.2 and 4.0 W m$^{-2}$, respectively). The RMSEs (rRMSEs) of clear-sky DLR estimated by the parameterization models were approximately 14.8 W m$^{-2}$ (5.0 %), 14.4 W m$^{-2}$ (4.0 %), 13.0 W m$^{-2}$ (4.2 %), and 13.6 W m$^{-2}$ (6.2 %) in spring, summer, autumn, and winter, respectively; the counterparts of all-sky DLR were approximately 18.6 W m$^{-2}$ (6.3 %), 18.4 W m$^{-2}$ (4.9 %), 17.7 W m$^{-2}$ (6.1 %), and 15.6 W m$^{-2}$ (7.3 %) in spring, summer, and winter, respectively.

**Table 4.** Comparison between the measured DLR and those estimated using three models in four seasons under clear- and all-sky conditions.

| Season | Sky condition | Model | MBE (Wm$^{-2}$) | rMBE (%) | RMSE (Wm$^{-2}$) | rRMSE (%) | $r$ | Sample number |
|--------|---------------|-------|-----------------|----------|------------------|-----------|-----|---------------|
| Spring | Clear sky | Brunt | −4.6 | −1.5 | 14.8 | 4.9 | 0.954 | 445 |
|        |           | Weng | −4.3 | −1.4 | 14.3 | 4.8 | 0.956 | |
|        |           | This work | 4.7 | 1.6 | 15.2 | 5.1 | 0.955 | |
|        | All sky | Brunt | −8.3 | −2.8 | 18.7 | 6.4 | 0.948 | 4389 |
|        |           | Weng | −9.4 | −3.2 | 19.1 | 6.5 | 0.948 | |
|        |           | This work | −6.6 | −2.3 | 18.0 | 6.1 | 0.947 | |
| Summer | Clear sky | Brunt | −0.0 | −0.0 | 12.9 | 3.5 | 0.921 | 254 |
|        |           | Weng | −4.6 | −1.3 | 13.5 | 3.6 | 0.913 | |
|        |           | This work | 10.9 | 2.9 | 16.9 | 4.6 | 0.919 | |
|        | All sky | Brunt | −3.1 | −0.8 | 17.8 | 4.7 | 0.922 | 5036 |
|        |           | Weng | −7.0 | −1.9 | 20.5 | 5.4 | 0.902 | |
|        |           | This work | −0.1 | −0.0 | 17.0 | 4.5 | 0.924 | |
| Autumn | Clear sky | Brunt | −3.1 | −1.0 | 12.4 | 4.0 | 0.966 | 369 |
|        |           | Weng | −3.0 | −1.0 | 12.9 | 4.2 | 0.963 | |
|        |           | This work | 7.1 | 2.3 | 13.9 | 4.5 | 0.967 | |
|        | All sky | Brunt | 1.2 | 0.4 | 17.4 | 6.0 | 0.965 | 5106 |
|        |           | Weng | −0.8 | −0.3 | 17.5 | 6.0 | 0.965 | |
|        |           | This work | 4.0 | 1.4 | 18.2 | 6.3 | 0.963 | |
| Winter | Clear sky | Brunt | −6.6 | −3.0 | 14.2 | 6.5 | 0.940 | 638 |
|        |           | Weng | −7.1 | −3.3 | 14.2 | 6.5 | 0.946 | |
|        |           | This work | −1.8 | −0.8 | 12.5 | 5.7 | 0.947 | |
|        | All sky | Brunt | −1.9 | −0.9 | 15.7 | 7.3 | 0.941 | 6439 |

| | | | | | |
|---|---|---|---|---|---|
| Weng | −7.4 | −3.4 | 15.9 | 7.4 | 0.947 |
| This work | −3.4 | −1.6 | 15.2 | 7.1 | 0.942 |

**5 Discussion and conclusions**

To date, several empirical parameterization models to derive DLR from near-surface meteorological elements have been developed on the basis of field observations obtained at a few sites in China. In this study, we utilized a long-term dataset of hourly observations from seven CBSRN stations to recalculate the coefficients of the Brunt model, the Weng model, and a new parameterization model to estimate atmospheric effective emissivity and DLR under clear-sky and all-sky conditions. The main conclusions of this study are as follows.

Generally, all three parameterization models can reliably estimate emissivity and DLR under clear- and all-sky conditions, i.e., the MBEs between the measured clear-sky DLR and that estimated using the Brunt model, the Weng model, and new model developed in this study were −4.3, −5.1, and 3.7 W m$^{-2}$, respectively; for all-sky DLR, the corresponding MBEs were −2.8, −6.1, and −1.5 W m$^{-2}$, respectively.

On the basis of the long-term (2011–2022) hourly data measured at the seven CBSRN stations adopted in this study, it is reasonable to suggest that the parameterization models considered in this study have reasonable spatial representation and robustness.

This study used continuous hourly CF observations from the HY-WP1A, which remarkably improved the consideration of cloud effects on estimations of emissivity and DLR. For example, CF data with high temporal resolution can improve the accuracy in estimating emissivity and DLR, and provide the opportunity to study diurnal variations in DLR. On the other hand, the IR02 pyrgeometers currently used at the CBSRN stations should be replaced with more precise instruments such as the CGR4 pyrgeometer. The IR02 was found usually to produce irrational positive records of DLR under extreme cold and dry synoptic conditions, which might be caused by its large temperature dependency (within ±3 % under −10 to 40 ℃).

Owing to limited data observed at seven CBSRN in China are used in establishing the parameterizations, the formulae presented in this study are mainly suitable to retrieve the downward longwave radiation in China. In the future, more data obtained from worldwide radiation stations (e.g., the BSRN, SURFRAD, etc.) is expected to be involved to establish the parameterizations, which could improve their capability to retrieve downward longwave radiation over more diverse geographical and climatological regions around the world.

Though the dominant emitter of longwave radiation in the atmosphere is water vapor, other gases (e.g., $CO_2$ and $O_3$) and aerosols also emit longwave radiation. The effects of gases and aerosols on DLR, however, are not considered sufficiently in the parameterization models. It is expected that the influences of atmospheric components on the relationships between clear-sky emissivity and screen-level meteorological variables will be further explored by means of the comprehensive observations at Global Atmosphere Watch stations such as SDZ.

*Data availability*. The data used in this study can be provided by the corresponding author upon reasonable request.

*Author contributions.* JH and QC contributed to shaping the ideas and presenting research goal and constructive comments on the research. JY and WQ presented the construction of the paper. JY contributed to processing and analysis of the data as well as preparing the manuscript. WQ contributed the ideas, organized the research, performed the review, edited the manuscript, and provided the funding acquisitions.

*Competing interests.* The authors declare that they have no conflict of interest.

*Acknowledgements.* We greatly appreciate Qifeng Lu of the CEMC for providing valuable and stimulating comments. We also thank Na Liu of the National Meteorological Information Centre for her helpful suggestions on how to correctly use the radiation data and meteorological data. We wish to thank reviewers and the editor for their valuable insights that helped greatly strengthen the manuscript.

*Financial support.* This study was funded by the China Scholarship Council (No. 202205330024), National Key Research and Development Program of China (Grant No. 2017YFB0504002), National Science and Technology Infrastructure Platform Project (2017), and the Special Fund for Basic Scientific Research of Institute of Urban Meteorology (Grant No. IUMKY201735).

*Review statement.* This paper was edited by ××× and reviewed by ××× anonymous referees.

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
