# Peer review of "Parameterization of downward longwave radiation based on longterm baseline surface radiation measurements in China"

_Atmospheric Chemistry and Physics, 2022_

## Author Comment (AC1)

**Response to Reviewer 1 Comments**

**General comments:**

1. This study investigates different empirical parameterizations of the surface downward longwave radiation regarding the adequacy for their use in China. In addition, the authors develop a new empirical parameterization and perform a comprehensive evaluation using data from 7 stations from the Chinese Baseline Surface Radiation Network. The authors conclude that the parameterizations and associated coefficients they derive are suitable for the determination of downward longwave radiation over China. The paper is well written, fairly straightforward and clearly structured. The applied methods are sound.

Thank you for this comment. We are grateful for reviewer's constructive comments.

2. My main reservation with this study is the relatively limited applicability of its results, being only of use for the determination of downward longwave radiation in China. To make the paper more attractive for readers outside China and to increase its impact in the community, information on the applicability of these parameterizations outside China would be valuable. For example, the Baseline Surface Radiation Network (BSRN, www.bsrn.awi.de) with numerous worldwide distributed high quality radiation stations would provide a framework to test and calibrate these parameterizations under more diverse geographical and climatological conditions. This would then allow to investigate the more general applicability of the parameterizations and make it more interesting for the worldwide readership of ACP. While such a broader analysis might be challenging to achieve within the limited time of a revision phase, I would find it at least useful if the authors could add a discussion of the potential and limitations of these parameterizations for their use outside of China, in order to provide guidance to readers interested to apply them for the determination of downward longwave radiation in other parts of the world..

Thank you for this valuable comment. Exactly, if more data observed from radiation stations (e.g., the BSRN, SURFRAD, etc.) is involved in building the parameterizations, the formulae in this study would be more attractive and suitable to retrieve the downward longwave radiation over more diverse geographical and climatological conditions around the world.

Due to the limited time of a revision phase, this suggestion is planned to be carried out in our future work. Whereas, we added a paragraph in section "**Discussion and conclusions***" of revision: "Due to limited data obtained from CBSRN used in building the parameterizations, the formulae presented in this study are mainly suitable to retrieve the downward longwave radiation in China rather than outside area. In the future, more data obtained from worldwide radiation stations (e.g., the BSRN, SURFRAD, etc.) is expected to be involved to establish the parameterizations, which could improve their capability to retrieve downward longwave radiation over more diverse geographical and climatological regions around the world."

3. While I think the English is overall adequate, there are still numerous minor issues as indicated in the technical comments. As I certainly not have caught all of them, I encourage the authors to doublecheck the manuscript in this respect, ideally with the help of a native English speaker..

We have checked the manuscript again and corrected some grammar errors. In addition, a native English speaker expert from the **LucidPapers** was invited to revise our manuscript, which made an obvious improvement on grammar and words in the manuscript.

4. I recommend the acceptance of the manuscript after revisions as outlined above and below.

Thank you for your consideration. We expect the manuscript can be improved after this revision.

**Specific comments:**

1. L69ff In addition to the climate zones, it would also be interesting to know if the stations are located in an urban, industrial or rural setting. This can give some indications to what extent the measurements could be influenced by local anthropogenic pollution sources.

Thank you for this suggestion. We have supplemented some descriptions and the related reference in Section 2.1 in the revision: "Moreover, the CBSRN stations represent various land covers in China. For instance, MH is the northernmost meteorological station in China surrounded by forest, which is located in the northwestern suburbs of Mohe County, Heilongjiang province (Liu et al., 2018). XL lies on the central of Inner Mongolia, where the main land cover is steppe. YQ, located on the northern margin of the Tarim Basin, is one of the representative stations in the desert and Gobi in northwest of China. SDZ is located in the northern North China Plain and only a few small villages with a sparse population around it (Zhou et al., 2021). XC is located in the central Henan province, which is surrounded by a wheat field and become one of typical representative stations for farmland in China. WJ is located in Sichuan Basin, which represents a paddy field. As a part of the Dali National Climate Observatory near the Erhai Lake in Yunnan province, DL is a represent station for wetlands."

2. Table 1: it would be worthwhile to include in this table also the measurement period for each site.

Thank you for this important suggestion, which can exhibit the history of establishment of CBSRN. As a pilot station, the XL was founded by China Meteorological Administration (CMA) in 2007. After approximately six-year operation of XL, other sites (i.e., MH, YQ, SDZ, XC, WJ, and DL) were established by CMA in 2013. We have added a column named "Measure period" in Table 1 in the revision.

3. L103: are there also collocated upper air soundings (radiosondes) available at some of the stations? BSRN recommends the high quality radiation stations to include upper air soundings for the interpretation of the measured fluxes and testing of models.

It is true that the vertical detection or radiosonde can offer necessary data to help interpretation of the measured fluxes and testing of models at high quality radiation stations. Overall, the CBSRN stations involved in this study can be classified into three groups:

- Station with radiosonde (XL, WJ), in which two sounding observations at 0000 UTC and 1200 UTC are performed every day. Sometimes, two additional observations are carried out at 0600 UTC and 2000 UTC to meet special requirement.
- Station without radiosonde but with vertical detections (SDZ, DL). SDZ is not only a radiation station but also a Global Atmosphere Watch (GAW) station, in which several instruments (e.g. Microwave radiometer, Wind profile radar, Lidar, and Gradient observation tower) are adopted to detect the vertical structure of the atmosphere. DL is a National Climate Observatory, which has both tower observations and surface observation (meteorological elements, radiation, etc.).
- Station without radiosonde (MH, YQ, and XC). These stations are conventional weather stations in China, in which fundamental meteorological elements as well as radiation components are observed.

4. L130: I think the structure and formulation of the Brunt model and the Weng model should be explicitly described here or in the method section.

Thank you for this suggestion. We have added two sentences to explicitly describe the Brunt model and the Weng model in Section 4.1 in the revision: "The Brunt model is one of the earliest pronounced models, in which a simple formula connecting the downward radiation from the atmosphere, the total black-body radiation at temperature, and the vapour pressure (Brunt, 1932). The Weng model is one of the earliest parameterization presented to retrieve the DLR over China area from the atmospheric temperature and vapour pressure based on the experimental observation data on the Tibetan Plateau (Weng et al., 1993)."

5. L131:It would be good to describe precisely how the clear sky hourly data were identified as being clear sky.

Thank you for the comment. In the revision, We modified the description in L131: "In this study, the coefficients of the Brunt model and the Weng model were calibrated using the nonlinear curve fitting method with 12,368 hourly data pairs (DLR and e) under clear-sky condition (defined as the corresponding cloud fraction equal to zero) observed at seven CBSRN stations between January 2011 and December 2017."

6. L142ff: ok here come the formulations of the different parameterizations which I expected earlier on (comment L130). Maybe this part could be described in the method section in a paragraph describing the different parameterizations used in this study together with their formulas.

Thank you for the comment. This is revised same as comment L130.

7. L150: by eye it is hard to recognize much difference between the 2 models (red and black curve) for the dry conditions (e≤17.5 hPa), thus hard to fully appreciate the improved performance of the Weng model for dry conditons.

Thank you for your comment. Though small differences exist between the Brunt model and the Weng model in the case of the vapor pressure is less than 17.5 hPa, the red curve exhibits faintly higher than the black curve when the vapor pressure between 2.5 and 12.5 hPa.

8. L185ff: It is not clear to me how the structure of the parameterizations has been established. Why do they have precisely this form and not e.g. another one?

Thank you very much for your valuable comment. It is really a hard task to convert parameterization of the clear-sky DLR to that of all-sky DLR due to the determination the effect of cloudiness on the parameterizations. To illustrate clearly the background of the shapes of Eq. (5)–(7), a paragraph including the related references and formula is added in the Section 4.2 of the revision as follows:

Under all-sky conditions, the emission from clouds can supplement the radiation emitted by water vapor and other gases in the lower atmosphere. Therefore, the effective emissivity of the atmosphere is higher under all-sky condition compared to that under clear-sky condition (e.g., Li et al., 2017). Numerous formulae were presented to estimate the emissivity under all-sky condition based on the emissivity parameterization under clear-sky condition and cloud fraction (e.g., Maykut and Church, 1973; Crawford and Duchon, 1999; Duarte et al., 2006; Choi et al., 2008). The formula of Duarte et al. (2006) with an adjustment of atmospheric humidity was adopted in this study. For a site like Barrow, Alaska, where both the temperature and the partial pressure of water vapor are low during much of year, the effect of atmospheric humidity on emissivity under all-sky condition can be neglected (e.g., Maykut and Church, 1973). However, the temperature and atmospheric humidity over the CBSRN stations vary over a wide range during a year, the addition of moisture correction to the formula, thus, seems more reasonable. The structure of formula to estimate the emissivity under all-sky condition in this study as:

$$\varepsilon_{all} = \varepsilon_{clr}\left(1 - \alpha CF^{\beta}\right) + \gamma CF^{\delta} \emptyset^{\zeta}, \tag{5}$$

where $\varepsilon_{all}$ represent all-sky emissivity; $\varepsilon_{clr}$ is the clear-sky emissivity calculated using Eqs. (2)–(4); CF is the cloud fraction (0–1); $\emptyset$ is relative humidity (%); $\alpha, \beta, \gamma, \delta,$ and $\zeta$ are regression coefficients, which were derived using the dataset of observations recorded at seven CBSRN stations between January 2011 and December 2020.

9. L194ff: I understand the independent clear-sky dataset is independent in the temporal sense, i.e. the data stem from another period (from 2018 onward rather than before 2018), however still from the same stations. Two questions here: 1.) why not all 7 stations have been used, but only 4? 2.) Is there a chance to do a validation also at independent stations (not only independent times)? Basically one could use the entire worldwide BSRN dataset for this (see general comments). This would have the advantage that one could also get an idea on the performance of these parameterizations in other parts of the world under different regimes.

Thank you for your comments. To be frank, as a new generation of radiation observation networks in China, the CBSRN is not yet mature despite a lot of efforts have been put in instrument maintenance, regular calibration, and data quality control of the raw data. Particularly, some defects like instrument failure, program error, and light stoke exist in the process of observation, which damage the integrity of the data. Despite not all data from 7 stations have been used in this study, the long-term observation of radiation can provide sufficient independent samples to establish and validate the parameterizations.

Just as pointed out by the reviewer, the integrality and data quality of the database derived from the CBSRN is a key issue in the application. Recently, we have completed another study on quality-assured database of baseline surface radiation at SDZ, in which detailed description on the instruments, data quality control, dataset assessment, and database construction are described.

10. L202ff: Similar comment as above, why only 3 stations are used here for a validation and not all seven? Again also an evaluation with (spatially) independent stations would be interesting, ideally even outside China.

Thank you for your comments. Besides the reasons explained above, another consideration of data used in this study is the balance of samples, i.e., the number of data pairs used in establishing parameterization is about 2–3 times of the one used to validate the parameters, through which to guarantee the representative of the samples on the one hand and assure enough samples to validate the parameterizations on the other hand. It is a good idea to use radiation data outside China even all over the world to establish a robust parameterization to estimate DLR.

11. L207: Why should more samples necessarily help to reduce the MBEs.

Thank you for your question. As we all know random error always exist in measurements of both radiation components and meteorological elements. In general, the distribution of MBE of emissivity (or DLR) between measured and estimated obeys the normal distribution, i.e., most of the MBEs close to zero but a few extraordinary samples far from the zero. Moreover, the effects of extraordinary samples on the MBEs would be weakened with the increase of total samples due to its average compensation effects. Furthermore, the calculated MBE would be more robust if more samples are used during calculation due to the smoothing effect of the samples.

12. L239ff/Figure 5: I assume this validation uses hourly values? And uses measurements from all 7 stations? This should be mentioned in the text or the figure caption.

Thank you for your reminder. Yes, this validation uses hourly measurements of DLR at four stations (YQ, XL, SDZ, and XC) during 2018−2021 and at three stations (XL, SDZ, and XC) from January 2021 to April 2022 to validate the clear-sky and all-sky DLR estimations derived from three models, respectively. In the section 4.4 of the revision, we have modified the corresponding sentences in the revision.

**Technical comments:**

1. L29: add "e.g.," in front of the references, as there are many other and also earlier papers dealing with DLR.

Thank you. This is fixed in the revision.

2. L31: same comment, add "e.g.," in front of the references, as there are many other application paper of DLR. There are several other places in the manuscript where an "e.g.,"

in front of the reference would be appropriate, as other papers could equally well be cited. The authors may check on this throughout the manuscript.

Thank you. This and others are fixed in the revision.

3. L40: hereinafter refer to > hereafter referred to as.

Thank you. This and others are fixed in the revision.

4. L44: the presence of cloudS

Thank you. This is fixed in the revision.

5. L59: in terms of > based on

Thank you. This is fixed in the revision.

6. L63: to estimation of > for the estimation of

Thank you. This is fixed in the revision.

7. L90: influencing > influences

Thank you. This is fixed in the revision.

8. L106: strictly quality controlled > strict quality controls

Thank you. This is fixed in the revision.

9. L140: hereinafter refer to > hereafter referred to as

Thank you. This is fixed in the revision.

10. L155: to have basis in physics > to be based on physics

Thank you. This is fixed in the revision.

11. L166: Circles represent data pairs > Circles represent hourly data pairs

Thank you. This is fixed in the revision.

12. L170: are in consistent > are consistent (inconsistent has the opposite meaning!)

Thank you. This is fixed in the revision.

13. L217: can overestimate > overestimates

Thank you. This is fixed in the revision.

14. L241 & L250: and that > and the one

*Thank you. This is fixed in the revision.*

15. L252 & L253 & L254 & L255: could underestimate > tend to underestimate

*Thank you. This is fixed in the revision.*

16. L269: three parameterization models > all three parameterization models

*Thank you. This is fixed in the revision.*

17. L278: improve accuracy > improve the accuracy

*Thank you. This is fixed in the revision.*

18. L279: this sentence sounds awkward and needs reformulation

*Thank you. This sentence is deleted but a replacement is added in the revision: "Owing to limited data observed at seven CBSRN in China are used in establishing the parameterizations, the formulae presented in this study are mainly suitable to retrieve the downward longwave radiation in China."*

19. L280: to establish > to be established

*Thank you. Due to the sentence mentioned in L279 is deleted, this problem is also solved.*

20. L286: whereas > however

*Thank you. This is fixed in the revision.*

21. L289: station > stations

*Thank you. This is fixed in the revision.*

---

## Author Comment (AC2)

**Response to Reviewer 2 Comments**

**Scientific aspect:**

1. The search for empirical expressions for longwave downwelling irradiance for China is a welcome contribution. The measurement of longwave down welling irradiance has been rare worldwide, and especially in China. In this sense, this reviewer does not demand the expansion of the scope to large regions. To build an empirical relationship between the longwave down welling irradiance with widely available climatic elements requires accurate irradiance measurements. The required accuracy is made not only of an instrumental accuracy, but also of the traceability to the international standard. This latter point is very important for any long-term observations, and is the basis for the Baseline Surface Radiation Network. This development comes from a bitter experience to realize serious differences among the longwave calibration methods practiced by many contries. It has become necessary to establish the global standard in longwave calibration, which is materialized as the World Standard Group of pyrgeometers at the World Radiation Centre in Davos. Within the BSRN, there was only one Chinese station, Xianghe, which jointed the BSRN, more than 10 years later than other sites, and ceased to operate already in 2015. The continued functioning of this site was an international wish. It is not a constructive direction for each country to establish own baseline radiation network. If this is done, however, like Chinese Baseline Radiation Network (CBSRN), its traceability to the World Standard must be established. This point is missing in the presented paper, reducing the trustworthiness of the accuracy of the proposed equations. The current status of the BSRN is summarized in Driemel et al. 2018, Earth Syst. Sci. Data, 10, 1491-1501. Li et al., 2013 may present the information on the CBSRN, but this literature is not accessible for most readers. Its main content can be introduced in the paper.

Thank you very much for your valuable comments. It is really true that the DLR measured at the CBSRN should traceable to the World Standard in order to improve its reliability and comparability compared to measurements from other radiation networks (e.g., BSRN). Fortunately, the pyrgeometers used in the CBSRN are irregularly calibrated via comparison with the reference CGR4 of China Meteorological Administration (CMA).

Therefore, in section 2.2 of the revision, two sentences and related references were added: "To assure the DLR measured at CBSRN is traceable to the World Radiometric Reference like that observed at Baseline Surface Radiation Network (Driemel et al., 2018), the IR02 pyrgeometers used in this study were calibrated against the reference CGR4 pyrgeometer (Kipp & zonen, the Netherlands) of China Meteorological Administration (CMA). The CGR4 can be traced to the World Infrared Standard by participating in the International Pyrgeometer Comparison organised by Physikalisch-Meteorologisches Observatorium Davos and World Radiation Center (e.g., Gröbner et al., 2014; PMOD/WRC, 2022)."

2. The empirical relationships under the cloudless sky are quite straightforward, as the depth of the atmospheric emission effectively reaching the surface the surface is quite thin as the authors pointed out. The mathematical shapes adopted in this sort of calculation are usually grey body emission. There are, however, at least two original proposals, which are independent of the graybody preoccupation. These original proposals are made by

Swinbank's 1963 QJRMS and Ruckstuhl's 2007 JGR papers, Swinbank is quoted in the manuscript but his originality is not appreciated.

Thank you very much for your valuable comments. The content between Line 39-44 in the manuscript is modified in the revision as: "Following the pioneering work of Ångström, numerous investigators have presented empirical relationships between effective atmospheric emissivity (hereafter referred to as emissivity) under clear-sky conditions and vapor pressure (e) (e.g., Brunt, 1932; Weng et al., 1993; Niemelä et al., 2001). Nevertheless, for a limited isothermal atmosphere emissivity would be less than unity and independent of temperature only if the atmosphere were of a constant greyness. In the real atmosphere, emissivity must, in principle, be temperature (Ta) dependent (e.g., Swinbank, 1963; Idso and Jackson, 1969). Moreover, some investigators even pointed out that the empirical emissivity depends on the dewpoint temperature (Td) (e.g., Berdahl and Fromberg, 1982), e and Ta (e.g., Brutsaert, 1975; Satterlund, 1979; Idso, 1981; Iziomon et al., 2003), relative humidity (ø) and Ta (e.g., Carmona et al., 2014) or even the total amount of water vapor (e.g., Ruckstuhl et al., 2007)."

3. The attempt to expand the empirical relationships to all sky conditions causes a problem, mainly owing to the diversity of the clouds and the limit in our observational documentation. Nevertheless, one of the earliest proposals was attempted by H. M. Bolz (1949) in Zeitschr. Meteorol. This is a systematic introduction of the effect of clouds. Relating to this matter, the description from Line 182 to Line 192 must be reformulated. It is necessary to present how the authors consider the shapes of Equations (5), (6) and (7) are justified, and how each independent variables offer the targeted results. There seem to be a slight confusion in expressing Greek variables also.

Thank you very much for your valuable comment. It is really a hard task to convert parameterization of the clear-sky DLR to all-sky DLR due to the determination the effect of cloudiness on the parameterizations. To illustrate clearly the background of the shapes of Eq. (5)–(7) in the manuscript, a paragraph including the related references and formula is added in the Section 4.2 of the revision as follows:

Under all-sky conditions, the emission from clouds can supplement the radiation emitted by water vapor and other gases in the lower atmosphere. Therefore, the effective emissivity of the atmosphere is higher under all-sky condition compared to that under clear-sky condition (e.g., Li et al., 2017). Numerous formulae were presented to estimate the emissivity under all-sky condition based on the emissivity parameterization under clear-sky condition and cloud fraction (e.g., Maykut and Church, 1973; Crawford and Duchon, 1999; Duarte et al., 2006; Choi et al., 2008). The formula of Duarte et al. (2006) with an adjustment of atmospheric humidity was adopted in this study. For a site like Barrow, Alaska, where both the temperature and the partial pressure of water vapor are low during much of year, the effect of atmospheric humidity on emissivity under all-sky condition can be neglected (e.g., Maykut and Church, 1973). However, the temperature and atmospheric humidity over the CBSRN stations vary over a wide range during a year, the addition of moisture correction to the formula, thus, seems more reasonable. The structure of formula to estimate the emissivity under all-sky condition in this study as:

$$\varepsilon_{all} = \varepsilon_{clr}\left(1 - \alpha CF^{\beta}\right) + \gamma CF^{\delta} \emptyset^{\zeta}, \tag{5}$$

where $\varepsilon_{all}$ represent all-sky emissivity; $\varepsilon_{clr}$ is the clear-sky emissivity calculated using Eqs. (2)–(4); CF is the cloud fraction (0–1); $\emptyset$ is relative humidity (%); $\alpha, \beta, \gamma, \delta,$ and $\zeta$ are regression coefficients, which were derived using the dataset of observations recorded at seven CBSRN stations between January 2011 and December 2020.

**Formalities:**

1. Generally, quoting earlier works for substantiating the point in the paper must be done carefully. Just quoting many papers does not support the point authors wish to make. As an example, let's take the first two sentences in the introduction, Line 28 to 32. The importance of the longwave downwelling radiation was not realized only in 1994 or 2020. These papers are rather recent papers in this subject. This reviewer suggests the authors to quote the first and most original paper on the subject and then several recent and best papers. Not all papers quoted in these lines do not necessarily represent the best knowledge of the present time.

Thank you for your comments, which urge us to pay more attention to the reference quoting in our future work. In this revision, We have added some important literature such as Idso and Jackson (1969), Wild and Cechet (2002) in the first paragraph of the Introduction. We also delete some literature that related less the object. Furthermore, we modified the unclear sentence in the Introduction, which responses in the comment 2 of "Scientific aspect", to improve the accuracy of expression in the Introduction.

2. Introduction can be shorter. Numerical detail can be summarily presented in Conclusions.

Thank you very much for your suggestion. As the main meaning of the last sentence in the **Introduction** is expressed in the **Discussion and conclusions**, it is deleted in the revision. ("This study represents an advance in comparison with previous work in terms of the following aspects: First, it not only recalculated the regression coefficients of the Brunt and Weng models, but also developed a new parametric formula suited for the estimation of DLR over China; 2) the hourly cloud fraction (CF) measured by a HY-WP1A Intelligent Weather Observation System was incorporated to considerably improve the handling of cloud effects in DLR retrieval under all-sky conditions; and 3) the spatial representativeness of the parameterization models over China was improved through use of measurements from the seven CBSRN stations in China"). After the modification, the **Introduction** in the revision seems more precise.

3. Line 44 to 49: in this discussion, Bolz's and Ruckstuhl's works can make a constructive contribution.

Thank you for your recommendation. Indeed, the work of Ruckstuhl (2007) gives us a lot of help and inspiration, which would not only improve our understanding in this study but also invoke our interesting of the greenhouse effects on the relationship between the DLR and ($T_a$, RH) in the future work.

4. Line 180, Table 2: Under Column Network/Site, Line Swinbank (1963), H. M. A. S. Diamantina in the Indian Ocean should be added to Aspendale and Kerang. The observation on board Diamantina over the Indian Ocean provided the measurement in very high humidity, and played an important role in generalizing Swinbank equation.

Thank you for your suggestion. This is fixed in the revision.

5. Line 185-191: There are confusions in Greek letters.

Thank you for your suggestion. This is fixed in the revision.

6. The analyses in Line 193 Section 4.3, and Line 236 Section 4.4 are well done and very useful.

Thank you for your approval.

7. Line 273-275: The seven CBSRN sites are all confined in the continental interior regions and do not represent the climate of the maritime regions. This bias should be considered.

Thank you for your comment.

Just as mentioned by the reviewer, within the BSRN, there was only one Chinese station, Xianghe, which ceased to operate in 2015. Fortunately, CMA has established the CBSRN in 2013, which consists of seven sites and plan to add other sites in the future. The CBSRN could fill up the deficiency of BSRN over China or even Eurasia to some extent. However, to be frank, as a new generation of radiation observation networks in China, the CBSRN is not yet mature despite a lot of efforts have been put in instrument maintenance, regular calibration, and data quality control of the raw data.

Due to the limited time of a revision phase, this suggestion is planned to be carried out in future work. Whereas, we added a paragraph in section "**Discussion and conclusions**" of revision: "Due to limited data obtained from CBSRN used in building the parameterizations, the formulae presented in this study are mainly suitable to retrieve the downward longwave radiation in China rather than outside area. In the future, more data obtained from worldwide radiation stations (e.g., the BSRN, SURFRAD, etc.) is expected to be involved to establish the parameterizations, which could improve their capability to retrieve downward longwave radiation over more diverse geographical and climatological regions around the world."

8. L373-379: The order of references, Liu, M. Q. et al. and Li, M. Y., et al. should be reversed. Likewise, Line 381-384: Niemelä et al., and Monteith can be reversed

Thank you for this suggestion. The order of references mentioned above and others are rearranged according to the alphabet of the first author's name.